



# Rock glacier characteristics serve as an indirect record of multiple alpine glacier advances in Taylor Valley, Antarctica

Kelsey Winsor[1*], Kate M. Swanger[1], Esther Babcock[2], Rachel D. Valletta[3&], James L. Dickson[4#]

[1]Department of Environmental, Earth and Atmospheric Sciences, University of Massachusetts Lowell, Lowell, 01854, United States of America

[2] Logic Geophysics & Analytics LLC, Anchorage, 99508, United States of America

[3]Department of Earth and Environmental Sciences, University of Pennsylvania, Philadelphia, 19104, United States of America

[4]Department of Earth, Environmental and Planetary Sciences, Brown University, Providence, 02912, United States of America

[*]Now at: School of Earth and Sustainability, Northern Arizona University, Flagstaff, 86005, United States of America

[&]Now at: Franklin Institute, Philadelphia, PA, 19103, United States of America

[#]Now at: Division of Geological and Planetary Sciences, California Institute of Technology, Pasadena, 91125, United States of America

*Correspondence to*: Kate M. Swanger (Kate_Swanger@uml.edu)

**Abstract.** The geomorphic record indicates that alpine glaciers in the McMurdo Dry Valleys of southern Victoria Land, Antarctica appear to advance during interglacial periods in response to ice-free conditions in the Ross Sea. Few records of these advances are preserved and/or subaerially exposed, complicating the interpretations of regional glacier response to climate changes. Here, we present geophysical and geochemical analyses of a rock glacier that originates from icefalls fed by alpine Doran Glacier in central Taylor Valley. The rock glacier exhibits a trend of increased weathering of granitic clasts via ventifaction and grussification down-flow. Meltwater ponds on the rock glacier exhibit variable salinity that ranges from freshwater to higher than seawater, with the highest salinity pond near the rock glacier toe. Ground-penetrating radar analyses reveal the feature to possess a primarily clean ice interior, with layers of englacial debris. Stable isotopic data from three ice cores support a glacial origin for the ice within the rock glacier. These data suggest that the current morphology of the rock glacier is the result of multiple events of increased ice contribution caused by advances of Doran Glacier, which is the main source of the ice that cores the rock glacier. We therefore demonstrate the potential of ice-cored rock glaciers to record multiple advances and retreats of Dry Valley glaciers, permitting the interpretation of glacial responses to Pleistocene and Holocene climate change even where direct records are not present.



## 1 Introduction

Alpine glaciers in East Antarctica's McMurdo Dry Valleys (Fig. 1) are important components of the regional hydrologic cycle (Lyons et al., 1998). In some areas, recent warming has caused melting of both alpine glaciers and ground ice, suggesting that modern ice behavior in the Dry Valleys might reflect global temperature changes (Fountain et al., 2006;

Hoffman et al., 2016). Partnered with an increase in both stream incision rates and lake levels throughout Taylor and surrounding valleys, this activity has suggested an invigoration of modern Dry Valley hydrology (Fountain et al., 2014; Guglielmin and Cannone, 2012; Levy et al., 2013; Macdonell et al., 2013). However, the geologic record of Dry Valley alpine glacier behavior previous to the Holocene is generally limited to isolated moraines and proglacial lake deposits that indicate prior glacial advances (e.g., Higgins et al., 2000; Swanger et al. 2017).

Here, we examine a Taylor Valley rock glacier that extends from an icefall fed by alpine Doran Glacier of the Kukri Hills (Figs. 1 and 2). Rock glaciers (flowing mixtures of ice, rock and sediments) are common throughout the Dry Valleys and other ice-free regions in Antarctica (Fukui et al., 2008; Hassinger and Mayewski, 1983), however their age, origin and geomorphic significance remains poorly constrained in many locations. In general, rock glaciers can form through

periglacial processes (permafrost and ice-rich talus) and through glacial processes (burial of glacier ice by talus and alluvium) (Anderson et al., 2018; Clark et al., 1998; Hamilton and Whalley, 1995; Knight et al., 2019). Therefore, rock glaciers might provide information on past glacial fluctuations, permafrost development and degradation, and erosion rates.

At the study location in Taylor Valley, because ice is fed to the rock glacier from the lateral margin of Doran Glacier,

intervals of rock glacier growth and advance are limited by the extent of Doran Glacier and its ability to advance to the top of a bedrock cliff (Fig. 3). To identify discrete episodes of Doran Glacier advance, we examine the subsurface structure, stable isotopic and major ion chemistry of buried ice and meltwater ponds, and surface weathering characteristics of the associated rock glacier. This rock glacier provides a unique chronological marker because it records multiple generations of rock glacier advance, which in turn record multiple advances and retreats of its source glacier.

## 2 Geologic Setting

Taylor Valley is the most well studied area of the Dry Valleys, an arid region predominantly free of surface ice (Fig. 1). At the western end of the valley, Taylor Glacier flows from the Taylor Dome of the East Antarctic Ice Sheet, and to the east, the valley is bounded by the Ross Sea. Taylor Glacier terminates ~35 km from the Ross Sea coast, into the west lobe of Lake Bonney, a 40-m deep, stratified lake with a high salinity gradient (Poreda et al., 2004; Wagner et al., 2010). South of this

lake, a 15–20º slope rises ~200 m to a bedrock platform. Here, Sollas Glacier flows north for ~8 km from the Kukri Hills and spreads onto a low-angle bench, reaching a width of ~1400 m and terminating at ~440 m above sea level (m asl). To the east,



a steep-walled arête separates Sollas Glacier from the smaller Doran Glacier, which flows north, is ~3 km long and terminates at ~840 m asl (Figs. 1 and 2).

Along its lower eastern margin, the Sollas Glacier terminates against a small, ice-cored moraine that includes stratified, dipping sediments (Fig. 2). Adjacent to the eastern margin of Sollas Glacier lies a narrow gulley with unconsolidated debris and isolated large boulders. The east side of the gulley is diamict-covered bedrock, which forms a steep slope that rises above the Sollas Glacier surface (Higgins et al., 2000). At the top of this slope is the rock glacier. Because the ice-cored debris and exposed icefalls are geomorphically connected, both are included in the mapped extent for the rock glacier (Fig. 2). The rock glacier extends ~1.2 km from ~1000 to 550 m asl, beginning at the Doran Glacier icefalls. The surface of the uppermost ~600 m of the rock glacier is currently exposed ice. Loose ice, both with and without bubbles, and cobbles/boulders are present in this upper section of the rock glacier. The lowermost ~600 m of the rock glacier is debris covered (Fig. 2). The debris-covered section has a concentric lobate form with hummocks and multiple steps and risers, indicative of active flow. Surface debris cover is generally thin, ranging from several cm in the upper portion of the rock glacier to > 30 cm near the toe (sediment cover thickens down-flow). Contraction cracks occur throughout the feature, and in at least one location clean ice is exposed within a vertical crack, showing the sharp contact between the buried ice and the overlying sediments. Along the eastern margin of the upper rock glacier, a steep-walled ice-cored ridge (similar in morphology to an ice-cored moraine) extends up-flow for ~300 m, buttressing the eastern margin of the modern exposed ice (Figs. 2 and 3b). The height of the rock glacier could be estimated along the western margin where it terminates on an exposed bedrock surface. At this location, the rock glacier margin is 10–12 m in height.

Nearly vertical bedrock cliffs (~100 m in height) divide Doran Glacier from the rock glacier, and are the main source of rockfall material to the landform. This granitic bedrock cliff is ventifacted. This region of Taylor Valley is dotted with extinct Plio-Pleistocene cinder cones $^{40}Ar/^{39}Ar$ dated by Wilch et al. (1993). These cinder cones also serve as source material for talus near Sollas and Doran glaciers. To the southwest of the rock glacier is a very steep cinder cone talus slope dated to $2.19 \pm 0.04$ Ma (Fig. 3a), which, while topographically isolated from the rock glacier at present, may have sourced mafic cobbles to the rock glacier area in the past. A second cinder cone, dated to $3.57 \pm 0.14$ Ma, is located east of the rock glacier (Fig. 3a). North of the rock glacier, where the ground surface flattens to nearly horizontal, the ground cover is dominated by mafic cobbles. These cobbles are commonly polished, sculpted, and/or pitted and are often underlain by salt efflorescences that can reach over 5 mm in thickness. In addition to the cinder cones, several mafic dikes visibly cross-cut the granitic bedrock of the arête east of Doran Glacier (Fig. 3a). Just to the northeast of the rock glacier is a grussified granitic boulder field with significant ventifaction, and where a mummified seal has been partially swallowed by a contraction crack.





## 2.1 Glacial History of Taylor Valley

Geomorphic evidence for Taylor Glacier advances occur on the Taylor Valley walls and floor. Recent advances of Dry Valley outlet and alpine glaciers are correlated with the presence of open water in the Ross Sea, which occurs during interglacial periods when the Ross Ice Shelf retreats (Naish et al., 2009). Thus, the outlet Taylor Glacier, in addition to the

smaller alpine glaciers appear to reach relative maxima during warm intervals due to higher precipitation and/or lower ablation rates (Higgins et al., 2000; Hoffman et al., 2016; Marchant et al., 1994; Swanger et al., 2017). Algal carbonates originating in proglacial lakes suggest that Taylor Glacier advanced during Marine Isotope Stage (MIS) 5, 7, 9 and possibly 11 (Higgins et al., 2000). The silt- and clay-rich Bonney drift marks the Taylor Glacier margin maximum during the last interglacial, and is found up to ~300 m asl (~140 m below the Sollas Glacier toe) (Higgins et al., 2000). Using cosmogenic

[3]He exposure dating, Swanger et al. (2017) correlated the outermost moraine associated with Stocking Glacier (17 km west of Sollas Glacier) to an advance during MIS 11 (Fig. 1). High-elevation deposits of Taylor and Ferrar outlet glaciers have also been dated to approximately 3–4 Ma (Staiger et al., 2006; Swanger et al., 2011). Many other geologic records of local ice advance and retreat are restricted to the last 20,000 years and show Holocene advances, that in many cases, surpass glacial extents in MIS 2 (Christ and Bierman, 2019; Hall et al., 2000).

## 3 Methods

### 3.1 Ice Coring and Sampling

We extracted seven shallow ice cores of buried ice and frozen meltwater ponds along a 500-m long transect down the centerline of the rock glacier in early November 2015. Three of the cores (SLI-15-01, -04 and -05) were from buried ice and measured 1.5–2.3 m in length (Figs. 2, 4, and Table 1). The remaining four cores (SLI-15-02, -03, -06, and -07) were taken

from two frozen ponds on the surface of the rock glacier, and each measured <1.5 m. Core -03 was a duplicate of -02, and core -07 was a duplicate of -06. The depth of coring was limited by debris and equipment. Ice cores were taken with a 7.6-cm diameter SIPRE hand auger. In addition to the seven ice cores, fifteen hand samples were gathered from the upper 15 cm of the buried ice, the exposed icefalls and surface ponds (Tables 1 and 2). Samples were packed frozen in sterile Whirlpak bags. From McMurdo Station, samples were shipped frozen to the University of Massachusetts Lowell (if sediment-rich) or

the National Ice Core Laboratory (NICL) in Lakewood, Colorado (if relatively clean). Subsamples for isotopic and elemental analyses were collected using the facilities at NICL, where laboratory storage freezers maintained sample temperatures below -20°C.

### 3.2 Major Ion Analyses

We sampled ten small, frozen meltwater ponds between the upper rock glacier and its toe (Fig. 5). Table 2 lists the

geographic location of each sampled pond and the maximum dimensions of the ponds. Based on repeat aerial photography



from 1956 to the present, all of the surface ponds on the rock glacier vary in size, generally reaching minima in the 1970s and 1980s. Five of the ponds (KWL-15-03, SLI-15-02, -06, -17, and -18) are persistent features in the aerial photographs, probably due to their relatively large sizes (Figs. 5c, f, g, and h). The remaining sampled ponds are more ephemeral (KWL-15-01, -02, SLI-15-10, -16, and -19). Frozen hand samples were collected using a clean, stainless steel ice pick or a trowel covered in a clean Whirlpak bag. Samples were stored frozen in HDPE bottles in the field, during transit, and in the laboratory. When ready for processing, samples were thawed and two aliquots were removed—one for cation and one for anion analyses. The aliquot allocated for cation analyses was acidified with ~0.5% trace metal grade HCl. Major ion analyses were performed at the University of Colorado Boulder, using ion chromatography for anion samples and an inductively coupled plasma optical emission spectrometer (ICP-OES) for cation samples. Species analyzed were: $Ca^{2+}$, $Cl^-$, $F^-$, Fe, $K^+$, $Mg^{2+}$, Mn, $Na^+$, $NO_3^-$, $PO_4^{3-}$, and Si.

### 3.3 Stable Isotope Analyses

All ice samples were kept frozen until melted in airtight containers and then stored in 4 mL HDPE bottles. All bottles were pre-rinsed in a 10% ACS grade nitric acid bath and triple rinsed with deionized water before packing, when the samples were sealed and refrozen until analyzed. Stable isotopic analyses (n=78) were conducted using an isotope ratio mass spectrometer at the Boston University Stable Isotope Laboratory. Analyses for $\delta^{18}O$ were performed via $CO_2$ equilibration, and deuterium analyses were performed via pyrolysis using a GVI ChromeHD™ system. Analytical precision for both measurements is typically ± 0.1‰. Isotope values are presented as per mil (‰) relative to Vienna Standard Mean Ocean Water (VSMOW).

### 3.4 Ground-penetrating Radar

#### 3.4.1 Data Collection

We used a Geophysical Survey Systems, Inc. (GSSI) ground-penetrating radar (GPR) system with a SIR-3000 controller to image the subsurface at the field site. We surveyed one longitudinal transect (approximately 500 m), composed of multiple lines, down the rock glacier from the highest ice-core site to the lowest (Fig. 2). We also collected one west-east transverse GPR line (approximately 120 m) at the boundary between debris-covered ice and exposed ice. The line extends across ~20 m of debris-covered ice and then onto the modern icefalls, stopping at the base of the east-lateral ice-cored ridge (Fig. 2). We collected data with 200-MHz and 400-Mz shielded antenna units, which are both configured in a single housing unit. For 200 MHz antenna, we took 2048 samples per trace, with a time window of 350–700 ns, and trace spacing of 0.05 m. The same sample parameters for the 400 MHz antenna were 4096 samples per trace, 150–500 ns window, and 0.1 m trace spacing. For both antennae, stacking was four and trace gain was -20 dB.



Typical surface roughness of rock glaciers prevents effective ground-coupling of the radar system. This reduced coupling degrades the depth of the signal penetration and the quality of the radargram. To combat these problems, before each line collection we removed surface clasts >0.1 m$^3$ from the GPR travel path and smoothed the surface sediment where possible. However, in places where clasts were frozen to the surface or too large to move, data quality was unavoidably degraded. We

surveyed the start and end point of each GPR line with a handheld GPS receiver. We also marked the locations in the data files that corresponded to coring sites. Finally, after each line collection, we surveyed the slope along the line with a Brunton Pocket Transit and replaced all boulders.

### 3.4.2 Data Processing

Initial data processing steps were dewow to remove very low-frequency components from the recorded data (Annan, 2005),

distance normalization to correct for odometer errors (Shean and Marchant, 2010), and time-zero correction to the first peak of the direct arrival (Yelf and Yelf, 2007). We also filtered the data with a bandpass frequency filter (200 MHz: 50-100-400-800 MHz; 400 MHz: 100-200-1000-2000 MHz) and then a background subtraction to enhance signal-to-noise ratio. We gained the data to enhance deeper reflections and cut the data to the useful time window. After these steps, analysis of diffraction hyperbolas provided radar-wave velocity estimates (Shean and Marchant, 2010). We used this velocity to apply

topographic migration and topographic corrections to each line using surface profiles from the Brunton measurements. All data processing occurred with Reflex-2D processing software (Sandmeier, 2008).

## 4 Results

### 4.1 Rock Weathering Trends

The most abundant lithology on the rock glacier surface is granite (Figs. 5 and 6). In the upper half of the rock glacier,

granitic clasts are primarily cobble- to boulder-size, subangular to very angular, and generally lacking in evidence of ventifaction (Figs. 5a, 5b, and 6a). Occasional large boulders exhibit ventifact pitting, although these may be relict features inherited from *in situ* weathering while exposed along the cliff below Doran Glacier (e.g., Fig. 3b). In some areas, packed, angular pebbles are present between larger clasts (Fig. 5b). On the distal sector of the rock glacier, the character of surface clasts changes markedly. Granitic boulders are rounded and in some cases ventifaction and/or salt weathering has caused the

development of surface pitting (Fig. 6b). The ground cover is grus, dominated by gravels and coarse sands (Figs. 5d, 5h and 6c). Where snowdrifts are protected by boulders, salt efflorescences and crusts can be found. Although cinder cone clasts are rare on the upper rock glacier, the lower rock glacier hosts patches of this material (Figs. 5i and 6c), particularly in depressions that hold small ponds or appear to have held them in the past (evidenced by salt deposits). Pebble- to cobble-sized cinder cone clasts cover these depressions in a loose, poorly developed desert pavement. Occasional boulders of cinder

cone material are also present on the lower rock glacier (Figs. 5g).





### 4.2 Major Ion Concentrations

Meltwater pond major ion concentrations range from those of freshwater to higher than those of seawater (Fig. 7 and Table 3). These results are compared to the average ion concentrations of seawater as described in Holland (1984). Highest major ion concentrations are found in the KWL-15-02, -03, and SLI-15-16 ponds, whereas the lowest concentrations are present in SLI-15-02, -18, and -19 (Fig. 8). Ratios of $Ca^{2+}:Mg^{2+}$, $Ca^{2+}:(K^+ + Na^+)$, $Ca^{2+}:Cl^-$ and $SO_4^{2-}:Cl^-$ decrease with increasing major ion concentrations (Fig. 7). The three ponds with the highest concentrations of $Cl^-$ also exhibit low ratios of $(K^+ + Na^+):Cl^-$ relative to the other ponds. The concentrations of $Cl^-$ and $Na^+$ in the most saline pond, KWL-15-03, are similar to that of seawater (Fig. 7a). Major ion concentrations are not correlated with pond size. The three most saline ponds are located in the lower rock glacier region, however, low salinity ponds also occur in the lower rock glacier. Major ions from samples collected from the lower margins of Doran Glacier and Sollas Glacier show low total ion concentrations (Table 3).

### 4.3 Stable Isotopes

Stable isotopic analyses were conducted on 54 samples from the buried ice (49 from the ice cores and five from the hand samples), 15 samples from surface ponds, five samples from the exposed icefalls, and four samples from Sollas and Doran glaciers. Most of the ice samples fall near or slightly below the local meteoric water line (LMWL) defined by Gooseff et al. (2006) (Fig. 9). The local alpine glaciers (Doran and Sollas) yield $\delta^{18}O$ values from -30 and -31‰, falling near the LMWL. Stable isotopic results from the three buried ice cores vary compared to the four glacier ice samples, but they are within the values observed from alpine glaciers in Taylor Valley (Gooseff et al., 2006). When plotted on a graph of $\delta^{18}O$ vs. $\delta D$, all 49 samples from the three buried ice cores (SLI-15-01, -04, and -05) fall on a slope of 7.7 with an $R^2$ value of 0.99. Samples from meltwater ponds display a large isotopic variance, with $\delta^{18}O$ values ranging from -31 to -22‰. The pond samples fall below the LMWL on a slope of 6.0, with an $R^2$ value of 0.94.

### 4.4 GPR

Mean radar-wave velocity derived from analyses of diffraction hyperbolas throughout the data sets was approximately 0.15–0.16 m ns$^{-1}$. Migrations using this velocity successfully collapsed diffractions without introducing data artifacts (Figs. 10 and 11). However, the overlying debris layer of typically less than 0.2 m thickness likely has lower velocity. This layered velocity structure is a source of error in our processing and subsequent analysis. In spite of the problems encountered with ground-coupling due to surface roughness, effective depth of penetration after migration was 10–20 m in the 400-MHz data and >15 m in the 200-MHz data. The processed data exhibited coherent reflection events within 0.1–0.2 m of the surface and below (Fig. 10). These reflection events generally appear as flat-lying and dipping reflectors separating portions of the radargrams with comparatively low signal content (Fig. 10).



## 5 Discussion

### 5.1 Stable Isotopes and Buried Ice Origin

Stable isotopic data from the buried ice, pond, and alpine glaciers support a glacial origin for the ice that cores the rock glacier, with some meltwater infiltration and reworking. Doran and Sollas Glaciers are isotopically similar and follow the general trend of Taylor Valley (more depleted isotopic values for inland glaciers compared to glaciers near the coast) (Gooseff et al., 2006). The modern glacier ice also falls around the median isotopic values for all samples from the rock glacier region. The pond ice is isotopically heavy with respect to oxygen and plots well below the LMWL, indicating that it has experienced greater isotopic fractionation due to evaporation compared to the buried ice and glacier ice.

At two sites at the boundary between the icefalls and the debris-covered ice, SLI-15-08 and -09 (Fig. 8), the sampled exposed ice was a mélange of bubble-rich ice chunks in a clear-ice matrix. At these locations, the bubble-rich ice falls on the LMWL isotopically, whereas the clear-ice matrices fall below (Fig. 9). The bubble-rich samples are likely preserved glacial ice. This interpretation is supported by the presence of fine spherical gas inclusions and snow accumulation stratigraphy. Conversely, the clear-ice matrices appear to be refrozen meltwater that has experienced minor evaporation fractionation.

### 5.2 Rock Glacier Subsurface Structure

Ground-penetrating radar along the rock glacier (Fig. 2) shows clear subsurface structure and clean ice (selected radargrams shown in Figs. 10 and 11). The presence of clean ice was verified via field excavation and was often present within 30 cm of the ground surface. The shallowest reflection event in the radar data corresponds to the interface between overlying unconsolidated sediment and the underlying clean ice containing scattered debris, as confirmed by excavation. Where the longitudinal GPR transect crossed ice-coring locations, we were able to correlate low GPR signal, or radar transparent regions (Brown et al., 2009), in the radargrams to the clean ice observed in the cores (Fig. 11). In the west-east (transverse) line across the boundary between the exposed icefalls and the buried ice (Figs. 2 and 10), we observe the same characteristically low-signal areas in the radargram interrupted by a prominent internal reflection event. This internal reflection event highlights the signal quality and radar transparency of the data above and below it. Additionally, the mean radar-wave velocity throughout our data set, obtained from diffraction-pattern analyses, is within 6% of that of clean, freshwater ice (0.167 m ns$^{-1}$). Finally, the depth of penetration for both frequencies employed is triple or greater than that normally seen when using these frequencies for rock or soil materials. This increased penetration is likely due to the low attenuation typically exhibited by radar waves traveling through cold ice (Arcone et al., 1995). Combined, these observations support our GPR-derived interpretation that the rock glacier is primarily ice-cored. Near the boundary between the exposed icefalls and the debris-covered ice, ice thickness is > 6 m (Fig. 10), and near the toe of the rock glacier, ice thickness is > 10 m in some locations (Fig. 11).



The prominent reflection events in the survey across the exposed icefalls are interpreted to be internal debris layers. On the transverse line (Fig. 10), the primary reflection event from the surface debris layer extends into the exposed icefalls and then dips upward toward the surface of the bounding ice-cored ridge (on the eastern rock glacier margin). The reflector maintains coherency especially in the first meter of descent away from the debris-covered ice (line position 20–30 m), supporting our

interpretation that it represents continuous internal debris. Thereafter, the layer maintains a consistent polarity indicative of the ice-debris-ice interface, although it becomes less coherent. At the approximately 50 m line position, another internal reflection event separates from the prominent debris layer and follows a similar (though less steep) dip, joining the primary upward-dipping debris layer around ~105 m line position. Below these prominent debris layers, more interpreted clean ice extends to at least the depth of signal penetration. Our observations also provide clear supporting evidence for a supraglacial

source of entrained debris that then travels englacially with rock-glacier flow. This pattern of repetitive internal and upward dipping reflectors is evocative of previous work on Dry Valley debris-covered glaciers, Mullins and Friedmann (Mackay et al., 2014).

Throughout the longitudinal transect, we observed similar internal reflectors and patterns, interpreted to be clean subsurface

ice with entrained debris bands (arrows, Fig. 11). Dominant dip of englacial debris layers is toward the toe of the rock glacier, corresponding to our interpretation of passive entrainment and subsequent redistribution of supraglacial debris. Where the longitudinal transect crosses surface ponds on the rock glacier, internal reflectors extend from the near-surface (~0.1 m) downward to form the base of the pond and presumably the interface between the melt ice and underlying bubble-rich glacier-derived ice (Shean and Marchant, 2010). We were unable to core below the ponds' basal layers. In pond SLI-15-

06 (Fig. 11b), two distinct debris layers bounding the pond run parallel to each other, while a third dips upward more abruptly. The lack of signal in the rest of the radargram near this location indicates the presence of clean ice below the surface debris layer, with a few areas having some entrained debris. Down rock glacier from the pond, it appears that this basal pond debris layer becomes further entrained and migrates down-flow.

The GPR data support multiple advances and retreats of the source for the rock glacier ice, Doran Glacier. In the transverse line (Fig. 10), the exposed ice from 30–115 m line position records the modern advance of Doran Glacier and the associated calving of ice from its lateral margin. The prominent subsurface debris layers below the modern ice likely record previous period(s) of Doran Glacier retreat. During retreat, the rock glacier loses its source for ice and a debris layer could accumulate at the rock-glacier surface via ablation and rockfall. Below the prominent debris layer, the low-signal zone (interpreted to be

more ice) likely records older advance(s) of Doran Glacier. Based on this interpretation, the transverse radargram records at least two recent advances of Doran Glacier (Fig. 10).



## 5.3 Surface Trends in Relative Age

The accumulation of major ions in meltwater ponds should be driven by two primary processes: snowfall and chemical weathering (Sun et al., 2015). Snowfall in the Dry Valleys contains aerosols of marine origin, which are a significant source of $Cl^-$ and $NO_3^-$ to the regional soils (Witherow et al., 2006). Although the surface sediments of the rock glacier do

experience summer thaw, the thaw season is generally short in this region (~34 degree days per year along Lake Bonney in the valley floor (Doran et al., 2002)). Compared to the coastal Dry Valleys, weathering is therefore less of a contributor to soil and meltwater major ions (Bao et al., 2008).

Based on the shape and position of meltwater pond KWL-15-02, it evolved from a snowbank (Fig. 5e), which supports the

interpretation that snowfall is a dominant contributor to the ponds. Some authors have reported increasing concentrations of the more soluble binary ions in downslope meltwater ponds due to the freeze-thaw-driven migration of those ions from upslope ponds (Healy et al., 2006; Lyons et al., 2012). However, we see a relative increase in the less soluble $Mg^{2+}$ in the lower meltwater ponds (Figs. 7b and 8), suggesting that the freeze-thaw process does not control the major ion concentration trend. Additionally, the hummocky, irregular surface topography of the rock glacier likely restricts connectivity between the

upper and lower rock glacier (Figs. 2 and 8).

Our major ion concentrations and cation ratios are dissimilar to those reported in meltwater ponds surrounding Marr Glacier, ~3 km northwest of the rock glacier (Fig. 1) (Lyons et al., 2012). This is likely due to the direct input of glacial meltwater to the Marr ponds, which possesses low concentrations of major ions as well as ratios driven strongly by snowfall. The rock

glacier ponds are more isolated and subject to evaporative concentration of ions. We suggest that the ponds in the upper glacier (SLI-15-02, -10 and KWL-15-01) have low total ion concentrations due to their relative youth and potential meltwater contribution from the Doran Glacier icefalls (Fig. 8). Ponds SLI-15-18 and -19 contain some of the lowest ion concentrations we measured, but occur near the rock glacier toe where the landform is older. However at this location, the surface morphology is distinct from the rest of the rock glacier: (1) the hummocks are lower relief and (2) the arcuate

transverse ridges are incised by fluvial channels that are oriented longitudinally to rock glacier flow. We were also unable to find clean subsurface ice in this lowest section of the rock glacier (based on five field excavations). Pond -19 is a small, ephemeral feature (Fig. 5i). Pond -18 is larger and long-lived, but in a shallow depression with a northern bounding bank that was < 50 cm above the modern pond surface. This configuration, along with the evidence for fluvial channeling, might allow pond water to flow out during high stands, thus limiting long-term concentration of ions in the depressions. Conversely,

pond KWL-15-03, which exhibits significantly higher ion concentrations than any of the other analyzed ponds, sits in one of the deepest depressions on the entire rock glacier, 2–5 m deep on all sides. At this location, the rock glacier is cored by > 10 m of clean ice (Fig. 11b), but sediment cover over the buried ice is > 30 cm thick (Fig. 4a), allowing for little to no melting of the buried ice as a source for pond water.



We estimate the maximum age of the highest salinity pond (KWL-15-03) using the average snowfall rate and average $Cl^-$ concentration in local annual snowfall, the measured $Cl^-$ concentration of the pond, and the modern volume of the pond (Fig. 5g). In this estimate, we assume that all pond $Cl^-$ originates from aerosols deposited by snow. We choose $Cl^-$ for this analysis because its distribution is controlled primarily by aerosol deposition (Keys and Williams, 1981; Witherow et al., 2006) and because it is the most abundant major ion in our ponds. A $Cl^-$ contribution from the chemical weathering of, in particular, cinder cone clasts (Bao et al., 2008), would decrease the estimated age. However, cinder cone clasts make up a small minority of surface clasts, especially in regions that lack desert pavement like the area around KWL-15-03. We use a modern volume of ~4,000 liters (~12 $m^2$ and ~50 cm deep) for pond KWL-15-03, and a snow accumulation area of ~600 $m^2$, which is the modern size of the depression that surrounds the pond, much larger than the pond itself. We use the depression size to include all snowfall that could currently contribute meltwater to the pond. We estimate $Cl^-$ flux to be ~150 µmol $m^{-2}$ $yr^{-1}$, which is the modern mean flux calculated from the four closest snow pits of Witherow et al. (2006) to Sollas Glacier. This yields a period of aerosol $Cl^-$ accumulation of ~40 kyr. This estimate is very rough, but it supports the interpretation that the lower rock glacier predates the Holocene, and is possibly MIS 3 or older.

## 5.4 Implications for McMurdo Dry Valleys Glaciers

Meltwater pond salinity and observations of surface weathering suggest that portions of the rock glacier have survived since before the Holocene. Distinct weathering regimes along the rock glacier surface support an episodic nature of rock glacier development, as do the dispersal of englacial debris and the arcuate surface ridges. If the rock glacier surface near pond KWL-15-03 is associated with MIS 3, it is possible that the rock glacier preserves evidence of advances from older interglacial intervals, such as MIS 5, and advances throughout the Holocene. Alpine glacier and rock glacier advance during MIS 3 would be consistent with records from alpine glaciers in the Denton Hills ~50 km south of Taylor Valley (Joy et al., 2017) and with evidence for open water conditions around East Antarctica during that period (Berg et al., 2016). Regardless of the exact age, the rock glacier preserves, in a single feature, a record of multiple alpine glacier advances for which the direct geomorphic evidence (such as moraines beyond the toe of Doran Glacier) is unavailable.

Taylor Valley glacial geomorphology shows evidence for advances of both outlet and alpine glaciers during MIS 5, 7, 9 and 11 (Higgins et al., 2000; Swanger et al., 2017), but provides little to no evidence for glacial fluctuations during the Last Glacial Maximum (LGM) and MIS 3. Additionally, even though Dry Valley glaciers appear to have advanced during the Holocene relative to the LGM, the timing and details of Holocene advance dynamics are unresolved (Hall and Denton, 2000). Therefore, if rock glaciers can preserve a record of glacier advance and retreat, these landforms can be used to constrain regional glacial fluctuations, especially where moraines are absent. In addition to the stated implications for the McMurdo Dry Valleys, our results have wider significance. Buried ice and rock glaciers are common throughout the ice-free

regions of Antarctica (Fukui et al., 2008; Hassinger and Mayewski, 1983; Bibby et al., 2016). Based on our results, it is likely that many ice-cored rock glaciers elsewhere in Antarctica record past glacial fluctuations, and can therefore be used to study climatic history during the Holocene and Pleistocene.

## 6 Conclusions

We use a combination of GPR, field observations, stable isotope and major ion analyses to characterize a rock glacier fed by the Doran Glacier icefalls in Taylor Valley. The rock glacier subsurface is primarily clean ice that is at least 10 m thick, with englacial debris bands that in places outcrop at the surface. Stable isotope values that are consistent with those of glacial ice, along with the presence of spherical gas inclusions, confirm that the buried ice within the rock glacier is glacial in origin. Surface weathering trends including the development of ventifacts, grus and minor desert pavement on the lower rock glacier suggest that the rock glacier toe is significantly older than the upper rock glacier. Major ion concentrations in meltwater ponds are variable, and consistent with aerosol deposition acting as the primary source for major ions. The total mass of $Cl^-$ in the most saline meltwater pond analyzed suggests an age of formation that predates the Holocene and the LGM. Our data support episodic advances of the rock glacier, which in turn record Doran Glacier margin fluctuations during late Pleistocene and Holocene time. This demonstrates the potential utility of Antarctic rock glaciers in preserving glacial behavior despite non-deposition of moraines and/or the destruction of older glacial landforms by younger advances.

## 7 Author Contributions

All authors participated in field observations, GPR surveying, and ice core collection. Swanger organized and led the field campaign. Winsor and Swanger collected soil and ice hand samples. Babcock directed GPR surveying and performed all GPR data processing. Winsor processed samples for major ion analyses. Winsor and Swanger processed samples for stable isotope analyses. Winsor, Swanger, and Babcock prepared the manuscript, created figures, and made tables.

## 8 Acknowledgments

We acknowledge the following funding source for support of this research: US National Science Foundation Office of Polar Programs grant 1341284. The authors would like to thank the McMurdo Station support staff, especially those at the Berg Field Center, Petroleum Helicopters International, and the Crary Laboratory, for assistance during the field season. Geoff Hargreaves and crew were of great assistance in subsampling ice cores at the National Ice Core Laboratory. Fred Luiszer (U Colorado–Boulder) analyzed samples for major ions and Robert Michener (Boston U) analyzed samples for stable isotopes. Useful discussions were had with Maciej Obryk.



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



Table 1. Rock glacier buried ice core and sample descriptions.

| Sample | Latitude, Longitude, Elevation[1] | Site description[2] | Ice core / sample description[3] |
|---|---|---|---|
| SLI-15-01 | 77.71722 S, 162.63182 E, 740 m asl | Debris ~10 cm thick, fresh granitic clasts, angular, little to no fines. | Ice core. 151 cm long. Clean ice, gas inclusions. |
| SLI-15-04 | 77.71628 S, 162.62518 E, 670 m asl | Debris ~20 cm thick, fresh and weathered granite, angular and rounded clasts, sand-rich matrix. | Ice core. 156 cm long. Clean ice, gas inclusions. Debris band at 130–135 cm depth. |
| SLI-15-05 | 77.71537 S, 162.61561 E, 610 m asl | Debris ~30 cm thick, fresh and weathered granite, rounded clasts, sand-rich matrix. | Ice core. 228 cm long. Clean ice, gas inclusions. Debris bands at 12, 70, 170, 190 cm depth. |
| SLI-15-08 | 77.71836 S, 162.63216 E, 780 m asl | Exposed ice at the boundary between icefalls and buried ice. | Hand sample. 5 cm depth. A: clean ice, gas inclusions. B and C: clean ice, no gas inclusions |
| SLI-15-09 | 77.71760 S, 162.63239 E, 760 m asl | Exposed ice at the boundary between icefalls and buried ice. | Hand sample. A: clean ice, gas inclusions. B: clean ice, no gas inclusions |
| SLI-15-11 | 77.71759 S, 162.62913 E, 730 m asl | Debris ~20 cm thick, fresh and weathered granitic clasts, sand matrix | Hand sample from an exposed contraction crack. Clean ice. 15 cm depth below ice-sediment boundary. |
| SLI-15-12 | 77.71699 S, 162.62979 E, 720 m asl | Debris ~10 cm thick, fresh angular granitic clasts, sands. | Hand sample from buried ice ~60 cm up-flow from pond SLI-15-02. Clean ice. 5 cm depth. |
| SLI-15-13 | 77.71694 S, 162.62891 E, 720 m asl | Debris ~5 cm thick, fresh angular granitic clasts, sands. | Hand sample from buried ice ~2 m down-flow from pond SLI-15-02. Clean ice. 5 cm depth. |
| SLI-15-14 | 77.71689 S, 162.62882 E, 720 m asl | Debris ~10 cm thick, fresh angular granitic clasts, sands. | Hand sample from buried ice ~10 m down-flow from pond SLI-15-02. Clean ice. 5 cm depth. |
| SLI-15-15 | 77.71676 S, 162.62831 E, 700 m asl | Debris ~5 cm thick, fresh and weathered angular granitic clasts, sands. | Hand sample from buried ice. Clean ice. 5 cm depth. |

[1] m asl = meters above sea level.
[2] Thickness and sedimentology of overlying debris.
[3] Length of ice core. Presence of gas inclusions and debris bands in buried ice. Depth of sampling.



Table 2. Location and description of sampled meltwater ponds

| Lake ID[1] | Maximum Lake Dimension | Latitude (S) | Longitude (E) | Elevation (m asl)[2] | Ground surface character |
|---|---|---|---|---|---|
| SLI-15-10 | 2 m | 77.71750 | 162.63167 | 720 | Fresh sands to boulders |
| SLI-15-02 | 16 m | 77.71697 | 162.62921 | 715 | Fresh and weathered boulders |
| KWL-15-01 | 4 m | 77.71660 | 162.62654 | 650 | Weathered boulders |
| SLI-15-16 | 1 m | 77.71623 | 162.62037 | 610 | Weathered boulders and grus |
| KWL-15-02 | 2 m | 77.71606 | 162.61954 | 595 | Weathered boulders and grus |
| SLI-15-17 | 25 m | 77.71611 | 162.61694 | 590 | Fresh boulders and grus |
| SLI-15-06 | 13 m | 77.71512 | 162.61662 | 610 | Weathered boulders and grus |
| KWL-15-03 | 8 m | 77.71559 | 162.61745 | 585 | Weathered boulders and grus |
| SLI-15-18 | 7 m | 77.71514 | 162.61486 | 565 | Weathered boulders and grus |
| SLI-15-19 | 3 m | 77.71510 | 162.61375 | 560 | Weathered boulders, grus and basalt cobbles |

[1] Lakes listed from south to north (upper rock glacier to terminus)
[2] meters above sea level





Table 3. Major ion results

| Sample ID[1] | Latitude (S) | Longitude (E) | Total major ions (ppm) |
|---|---|---|---|
| **Buried ice** | | | |
| SLI-15-01 | 77.71722 | 162.63182 | |
| 0 cm | | | 32.921 |
| 75 cm | | | 52.346 |
| 114 cm | | | 50.586 |
| 143 cm | | | 84.277 |
| SLI-15-04 | 77.71628 | 162.62518 | |
| 15 cm | | | 18.150 |
| 137 cm | | | 23.692 |
| 210 cm | | | 25.951 |
| SLI-15-05 | 77.71537 | 162.61561 | |
| 15 cm | | | 26.068 |
| 65 cm | | | 27.216 |
| 149 cm | | | 61.489 |
| SLI-15-08A | 77.71836 | 162.63216 | 9.726 |
| SLI-15-08B | | | 6.292 |
| SLI-15-08C | | | 4.754 |
| SLI-15-09A | 77.71760 | 162.63239 | 26.254 |
| SLI-15-09B | | | 10.743 |
| SLI-15-11 | 77.71759 | 162.62913 | 34.068 |
| SLI-15-12 | 77.71699 | 162.62979 | 29.410 |
| SLI-15-13 | 77.71694 | 162.62891 | 38.297 |
| SLI-15-14 | 77.71689 | 162.62882 | 17.197 |
| SLI-15-15 | 77.71676 | 162.62831 | 34.666 |
| **Meltwater ponds** | | | |
| SLI-15-02 | 77.71697 | 162.62921 | |
| 0 cm | | | 1.043 |
| 21 cm | | | 1.071 |
| 53 cm | | | 0.530 |
| 74 cm | | | 1.110 |
| SLI-15-06 | 77.71512 | 162.61662 | |
| 0 cm | | | 4.355 |
| 10 cm | | | 2.619 |
| KWL-15-01 | 77.71660 | 162.62654 | 11.585 |
| KWL-15-02 | 77.71606 | 162.61954 | 145.711 |
| KWL-15-03 | 77.71559 | 162.61745 | 47455.763 |
| SLI-15-10 | 77.71750 | 162.63167 | 13.769 |
| SLI-15-16 | 77.71623 | 162.62037 | 334.063 |
| SLI-15-17 | 77.71611 | 162.61694 | 17.199 |
| SLI-15-18 | 77.71514 | 162.61486 | 1.546 |
| SLI-15-19 | 77.71510 | 162.61375 | 3.791 |
| **Glaciers** | | | |
| Doran Gl. | 77.71481 | 162.65637 | 5.90 |
| Sollas Gl. | 77.71402 | 162.60377 | 18.06 |

[1] Depth below ice surface is listed in centimeters for ice cores. All other samples were gathered from 0–10 cm depth in buried ice or pond. Glacier samples taken from margins.



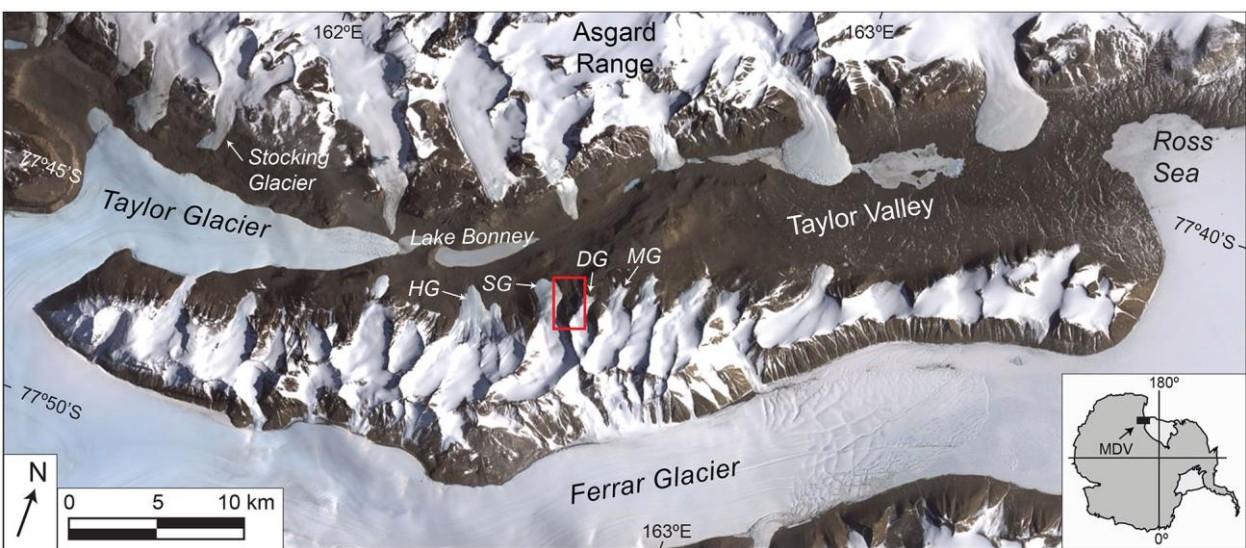

**Figure 1.** Satellite image of the McMurdo Dry Valleys (MDV), with inset indicating location in Antarctica. HG = Hughes Glacier, SG = Sollas Glacier, DG = Doran Glacier, and MG = Marr Glacier. Red rectangle shows location of rock glacier (Fig. 2). Public domain Landsat7 imagery courtesy of NASA Goddard Space Flight Center and U.S. Geological Survey.



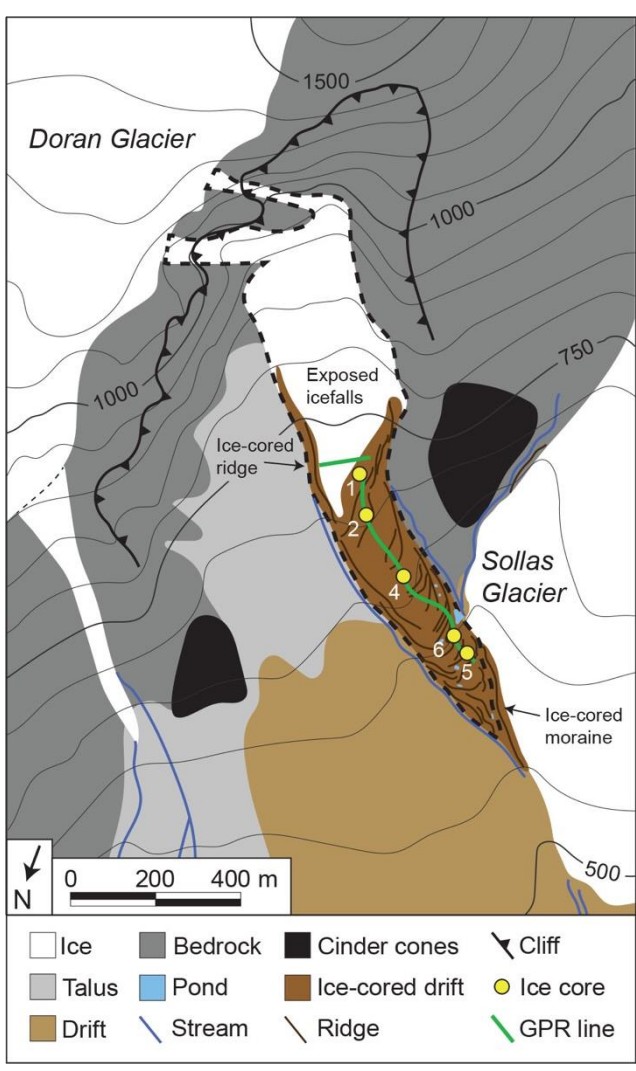

**Figure 2.** Geomorphic map of the modern rock glacier (outlined in the large black dashed line) and surrounding terrain. Contour interval: 50 m. Ice cores SLI-15-01, -04 and -05 are from buried ice (labelled 1, 4 and 5). SLI-15-02 and -03, and SLI-15-06 and -07 are duplicate cores taken from two large frozen ponds (only 2 and 6 labelled). All gathered ground penetrating radar (GPR) lines are shown in green, but

5    only two representative sections of the longitudinal line are discussed in the text.



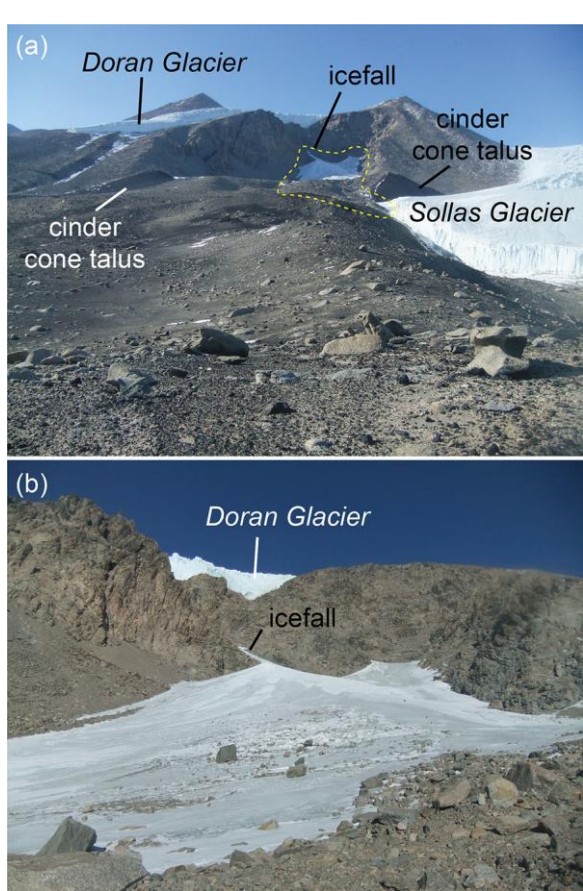

**Figure 3.** Icefall originating from Doran Glacier and accumulating to form the rock glacier. a) View from base of Sollas Glacier, with rock glacier visible within yellow dashed line. Note presence of cinder cone talus slopes. b) View from boundary between debris-covered ice and then modern exposed icefalls.



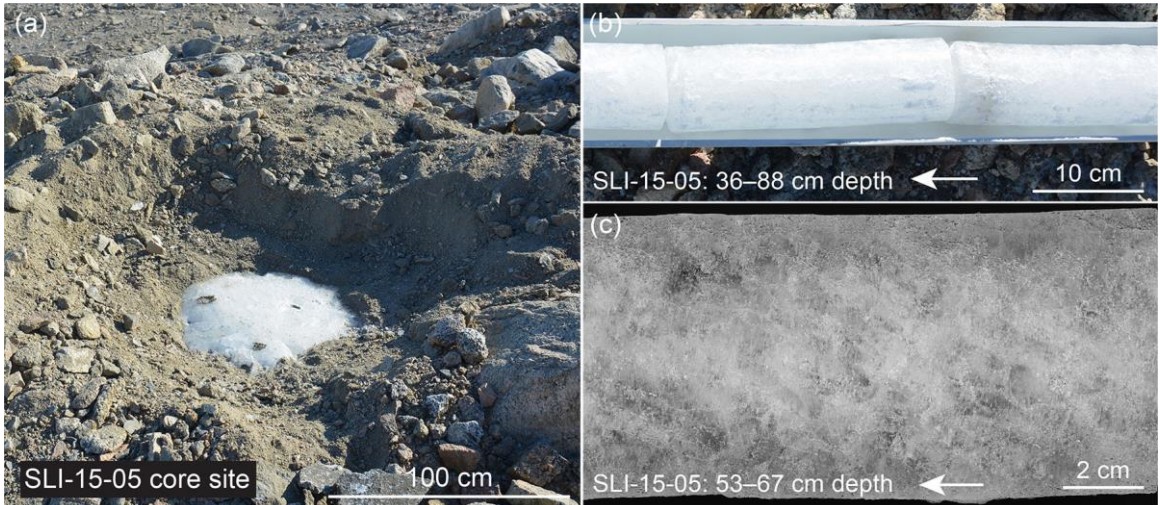

**Figure 4.** Buried ice core SLI-15-05. a) Excavation of ~30 cm of sand- and boulder-rich sediments above buried ice, exposing sharp contact between sediments and ice surface. b) Buried ice core pictured in the field, 36–88 cm depth below ice surface; up core is to the left (shown with arrow). c) Buried ice core (53–67 cm depth) after being cut vertically at NICL. At this depth, buried ice contained many gas inclusions and no sediments.



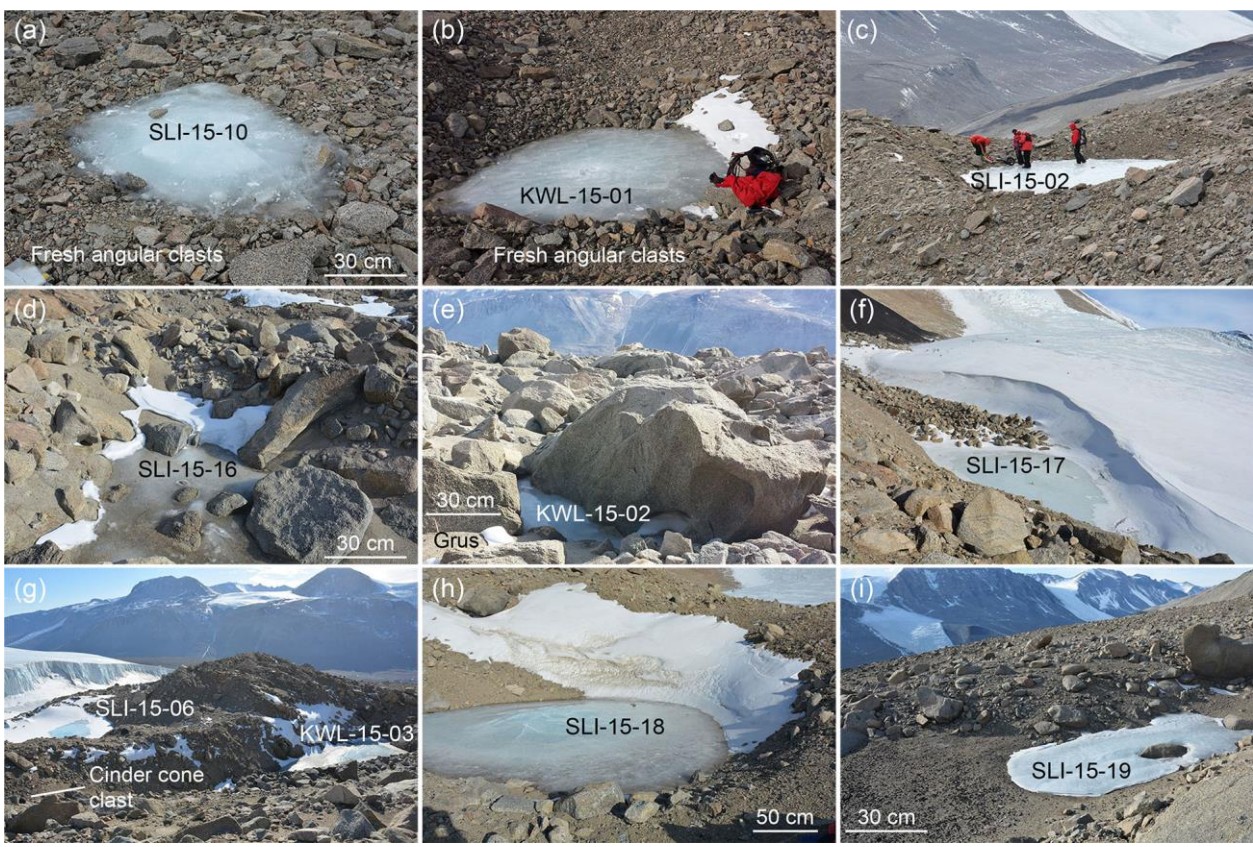

**Figure 5.** Field images of the ten sampled meltwater ponds along a longitudinal transect down the rock glacier. a) SLI-15-10 b) KWL-15-01, c) SLI-15-02, d) SLI-15-16, e) KWL-15-02, f) SLI-15-17, g) SLI-15-06 (on left) and KWL-15-03 (on right), h) SLI-15-18, i) SLI-15-19. Pond SLI-15-17 occurs between the rock glacier margin and Sollas Glacier. Note the fresh appearance of granites surrounding KWL-15-01 and SLI-15-10, and the grussified ground surface surrounding KWL-15-02 and SLI-15-16 and -18.



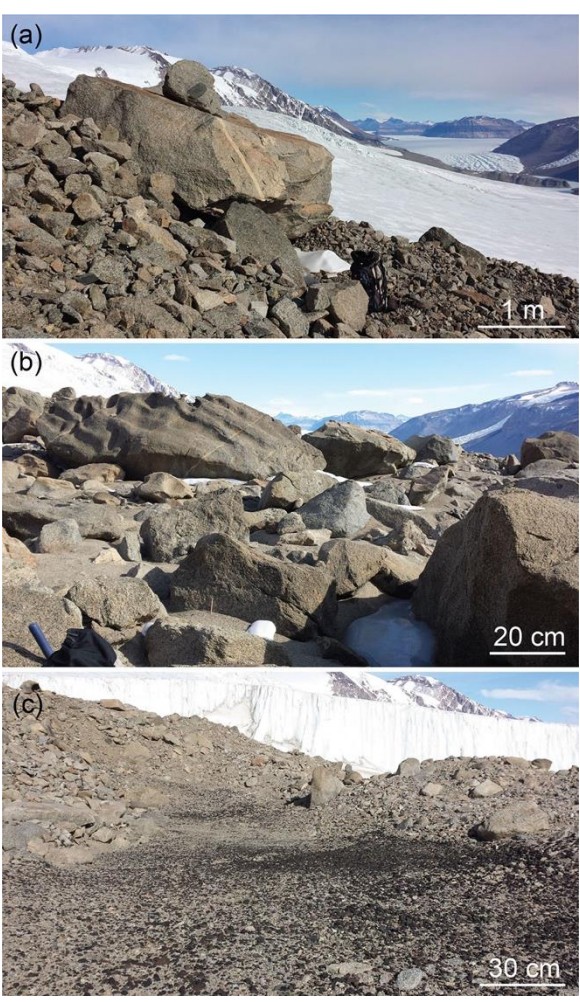

**Figure 6.** Surface lithology and weathering trend going down the rock glacier. a) Upper rock glacier, showing fresh, angular granitic boulders and cobbles. b) Central rock glacier, showing rounded and ventifacted granitic boulders and a granitic grus ground cover. c) Lower rock glacier, showing smaller clast sizes and poorly developed desert pavement in a small depression with mixed granitic and basalt ground cover.




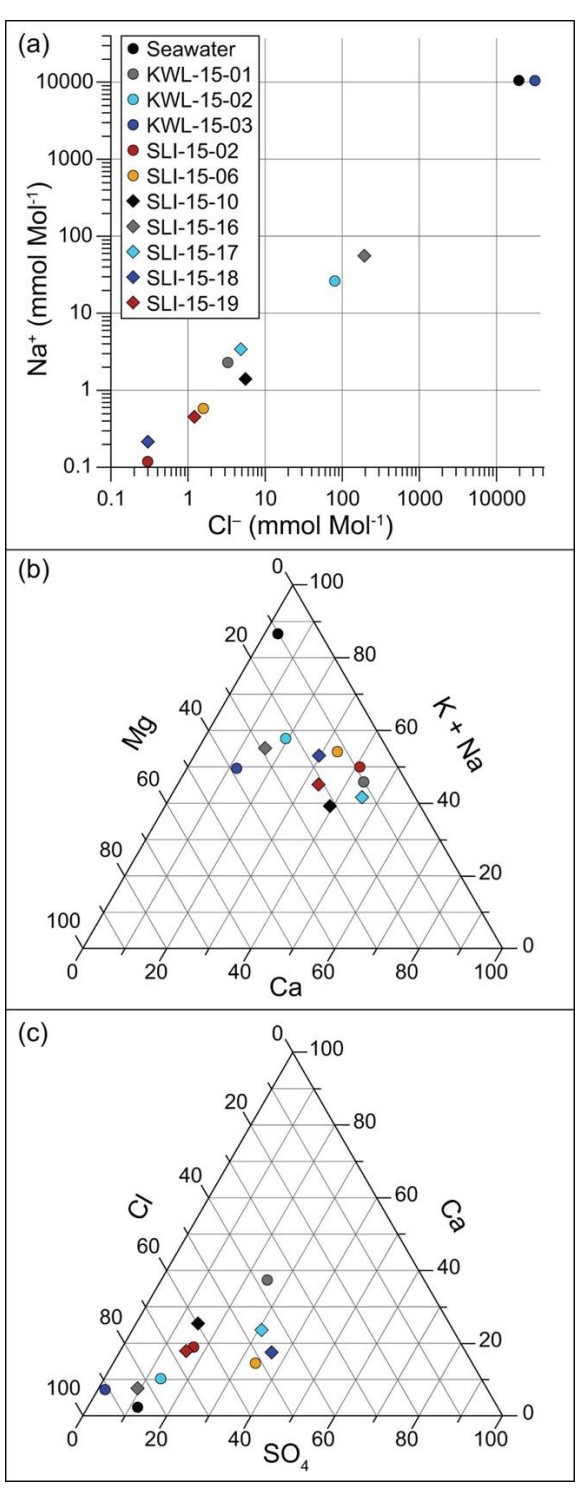

**Figure 7.** a) Scatter plot of $Na^+$ versus $Cl^-$ in ten meltwater ponds. Ternary diagrams of b) $Mg^{2+}$, $Ca^{2+}$, and $K^+ + Na^+$ and c) $Cl^-$, $SO_4^{2-}$, and $Ca^{2+}$ in meltwater ponds shown as percent by equivalent.

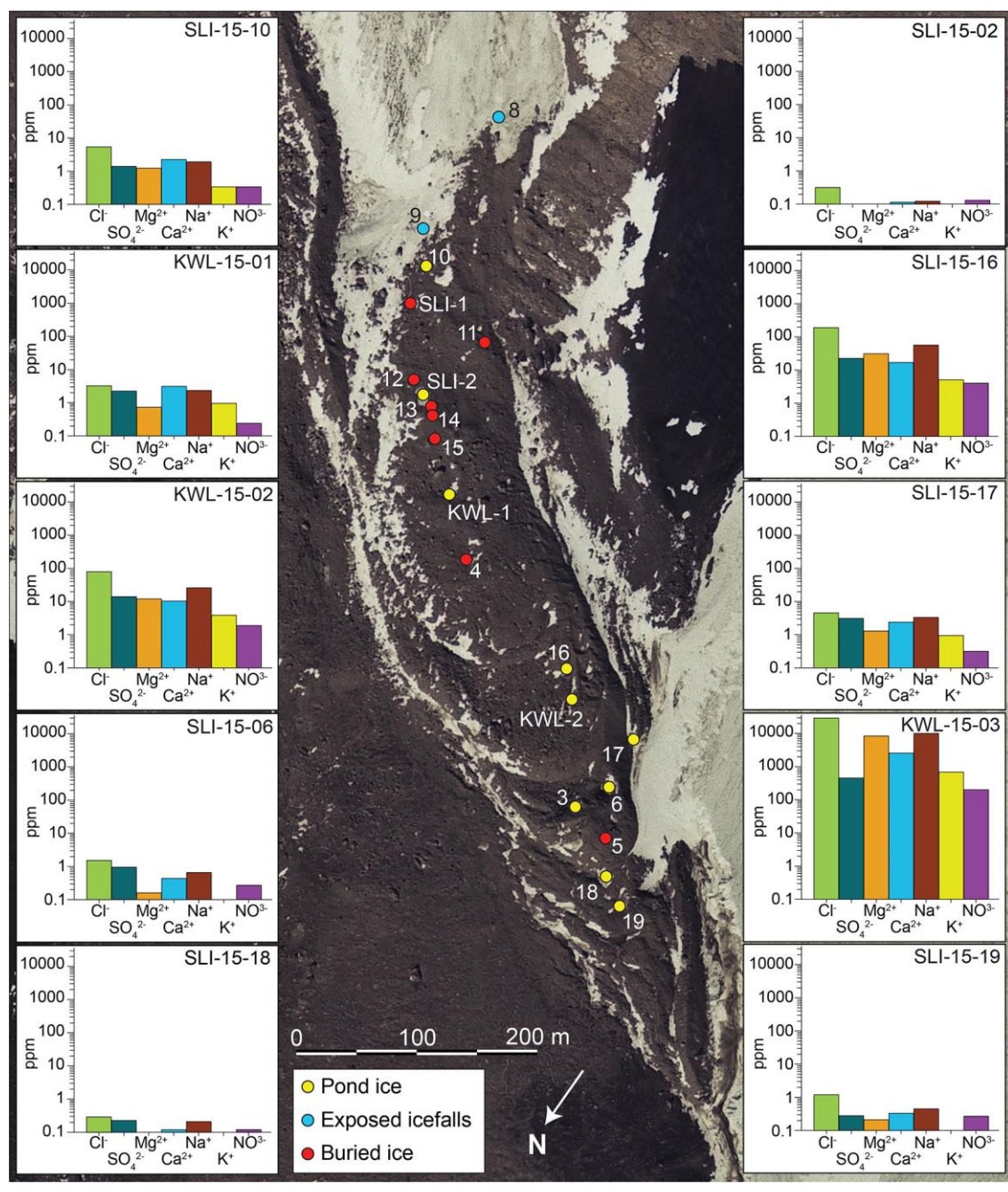

**Figure 8.** Aerial photo of the rock glacier with pond samples and buried ice samples. Cinder cone talus and the eastern margin of Sollas Glacier are both visible to the right (west) of the rock glacier. Bar charts show major ion concentrations from ten meltwater ponds (shown with yellow circles). All samples are labelled with their numbers only, except where the same sample numbers correspond to different samples (SLI-15-01 and -02 vs. KWL-15-01 and -02), labelled as SLI- and KWL-. Public domain aerial photograph from U.S. Geological Survey, TMA3084-V0132 taken in 1993.



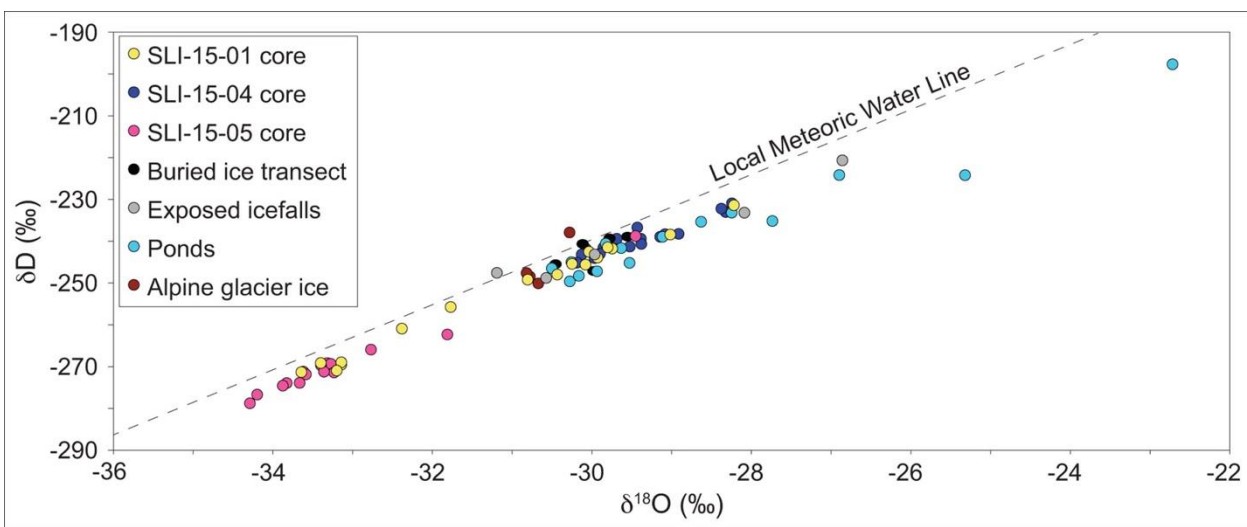

**Figure 9.** Plot of $\delta^{18}O$ vs. $\delta D$ of 59 samples from the rock glacier. Cores 01, 04 and 05 were gathered from buried ice along a longitudinal transect down the central flowline of the rock glacier. Buried ice transect refers to additional hand samples SLI-15-11 to -15 (see Fig. 8 for locations). Exposed icefalls were five samples from two locations (SLI-15-08 and -09; Fig. 8). Samples were also gathered from seven ponds along the rock glacier surface (SLI-15-02, -06, -10, -16, -17, -18 and -19; Fig. 8) and two local alpine glaciers, Sollas and Doran. The local meteoric water line is from Gooseff et al. (2006).
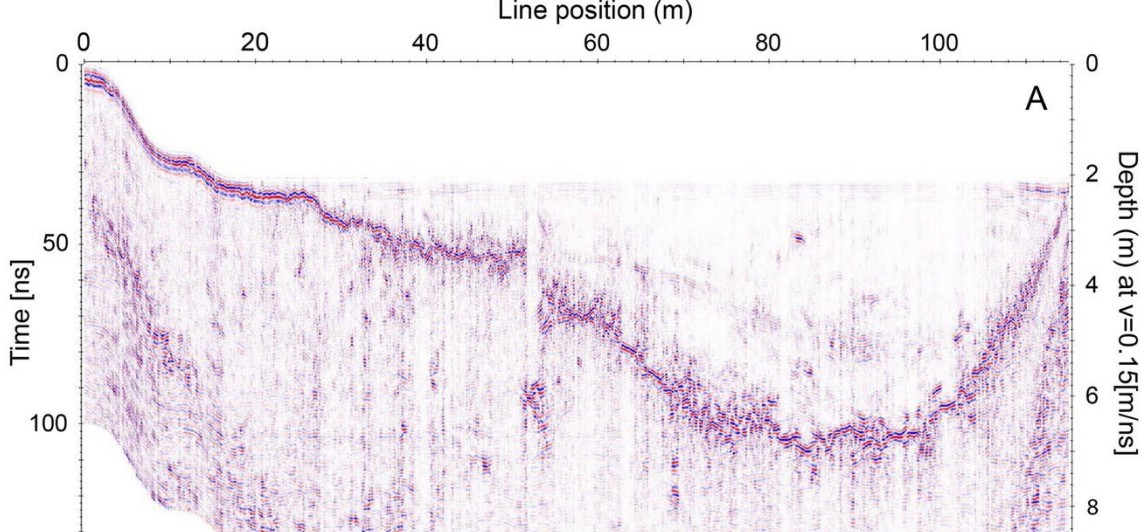

**Figure 10**. Processed ground-penetrating radar data for the transverse line (west to east) from the debris-covered rock glacier across the icefalls fed by Doran Glacier and ending at the east-lateral ice-cored ridge. Image after migration and topographic correction using a constant velocity of 0.16 m ns$^{-1}$. Note that the prominent internal reflection event starts as a near-surface reflection event at the beginning of the profile. At ~20 m line position, this primary interpreted surface-debris/internal-ice interface and dips down below the exposed ice before emerging at the ice-cored ridge (right side of profile).





**Figure 11.** Processed 400 MHz ground-penetrating radar data from the transect down the rock glacier. a) Mid-rock-glacier line showing interpreted internal debris and a mid-rock-glacier pond (~50–70 m; site of ice core sample SLI-15-02 and -03). b) Part of the transect crossing the frozen pond (sample site SLI-15-06 and -07) near the lowest buried ice-core site (at 110 m; SLI-15-05). Arrows point to reflection events interpreted to correlate with internal debris layers.