# Peer review of "Rock glacier characteristics serve as an indirect record of multiple alpine glacier advances in Taylor Valley, Antarctica"

_The Cryosphere, 2019_

## Referee Comment (RC1) · Anonymous Referee #1 · 26 Jun 2019

Summary: This article looks at a rock glacier in Taylor Valley and concludes that observed glacial successions are useful for studying past climates in the MDV. Overall, I think the article is well described and written, but I do have issues with their age estimate based on pond salts, as well as the description of the pond samples. Unfortunately, this is a key argument in the paper, so the lack of an age constraint requires changes to the discussion and conclusions. However, I don't think this greatly impacts the overall point.

Detailed comments:

Figure 2: I know that you can tell what is downhill from the contours, but at first glance

the figure seems turned around and was quite confusing, until I realized it was oriented differently from Fig. 1. I recommend including a big bold arrow and 'downhill' text, so that this is immediately clear.

Section 3.2: I'm a little confused as to exactly what samples were collected from the ponds, so more clarity is needed. Were all the ponds frozen during sampling? If so, where were the samples taken? From the upper surface, from the margins? I'm particularly curious because some of the pond samples have remarkably high salt concentrations, which is difficult to reconcile with samples of surface ice, which I'd expect to have low concentrations.

Table 3: The authors should fully list all of the ions analyzed and their concentrations, as well as the listed total salt content. Assuming that the isotopes were analyzed for the same samples, these values should also be listed. I also recommend adding in the elevation to the table for easy reference.

Section 4.3: It's worth mentioning here that samples falling below the local water line suggest either evaporation or sublimation.

Section 5.3: The major source of ions to these ponds is likely dissolution and aeolian transport of salts from nearby soils, as well as inputs of snow and ice melt, and probably a small component due to direct weathering or atmospheric inputs. See the following references on Taylor Valley salts:

Keys, J. R. H. and K. Williams (1981). "Origin of crystalline, cold desert salts in the McMurdo region, Antarctica." Geochimica et Cosmochimica Acta 45(12): 2299-2309.

Toner, J. D., et al. (2013). "Soluble salt accumulations in Taylor Valley, Antarctica: Implications for paleolakes and Ross Sea Ice Sheet dynamics." Journal of Geophysical Research 118(1): 198-215.

In general, this section needs improvement. Part of the difficulty I'm having with it is that I don't know exactly what was sampled, and if only a surface ice sample was

collected, then the measured chemistry would have little relation to the bulk pond composition/salinity. Also, the age analysis using Cl- accumulation is poorly justified. First, only a snowfall source is invoked, but as I mention, this is probably minor relative to Cl- fluxes from surrounding soils. Also, this assumes that the pond is a closed system, so that all the Cl- that goes in does not come out, but this seems unlikely for a pond perched on a valley slope. Throw in the aforementioned uncertainty about how representative the samples are. Basically, I don't think you can come up a with an age using snowfall. I recommend just removing this paragraph.

However, given that the age of these ponds is a major point in the article, this would involve some general restructuring of the later discussion and conclusions. On the whole, I find the argument for glacial successions is robust, but not the age estimate based on salt accumulation.

---

## Referee Comment (RC2) · Anonymous Referee #2 · 26 Aug 2019

This manuscript details a comprehensive field investigation of a very remote rock glacier in the dry valleys of Antarctica. The authors carefully assess the geomorphology of the rock glacier, examine its internal form using geophysics, and evaluate the chemical and isotopic make up of its ice and those of the meltwater ponds atop the glacier. All of this evidence suggests that the rock glacier has been stable for ∼40 ky, which is an impressive amount of time, given the controversy over the dynamics of a possible glacially-dammed lake at the mouth of the valley 5-10 ky ago. One thing that continues to challenge many of the researchers in the dry valleys is how such a lake could exist with so much ice in the area (locally) remaining intact. This set of evidence and conclusion is a terrific contribution to the dry valleys cryospheric literature

and provides an excellent example of the wealth of secrets kept in these rock glaciers, especially in Antarctica.

Minor suggestions: p. 2, line 21 – I suggest that 'subsurface structure' maybe be changed to 'internal structure', and that the authors add "(using GPR)" or something similar. The reason I mention this is that the GPR part of the Methods section sorta jumps out at the reader without much purpose. Hence, I propose putting some indication of the purpose back in the Introduction section.

p. 8, line 17 – 'ground surface' is a little confusing here. Do you mean where the ice (at depth) is in contact with the ground or do you mean from the surface of the rock glacier, which one must stand on. I suggest being more accurate with this and all other instances of 'ground surface' in the paper.

p. 10, lines 31-32 – the sentence "At this location, the rock glacier is cored by > 10 m of clean ice (Fig. 11b)," – should 'cored' be 'covered'? I didn't think that manual core collection got this deep... Oh wait, maybe you aren't meaning this to be the verb that has to do with collecting an ice sample... I'd suggest changing 'cored' to 'made up of' or something like that so that there's no confusion.

---

## Author Comment (AC1) · 17 Sep 2019

*We thank the anonymous reviewer for their time, expertise, and their insightful and helpful suggestions to make the manuscript more accessible, accurate, and impactful.*

Anonymous Referee #2
This manuscript details a comprehensive field investigation of a very remote rock glacier in the dry valleys of Antarctica. The authors carefully assess the geomorphology of the rock glacier, examine its internal form using geophysics, and evaluate the chemical and isotopic make up of its ice and those of the meltwater ponds atop the glacier. All of this evidence suggests that the rock glacier has been stable for ~40 ky, which is an impressive amount of time, given the controversy over the dynamics of a possible glacially-dammed lake at the mouth of the valley 5-10 ky ago. One thing that continues to challenge many of the researchers in the dry valleys is how such a lake could exist with so much ice in the area (locally) remaining intact. This set of evidence and conclusion is a terrific contribution to the dry valleys cryospheric literature and provides an excellent example of the wealth of secrets kept in these rock glaciers, especially in Antarctica.

*We thank the reviewer so much for these kind comments.*

Minor suggestions:
p. 2, line 21 – I suggest that 'subsurface structure' maybe be changed to 'internal structure', and that the authors add "(using GPR)" or something similar. The reason I mention this is that the GPR part of the Methods section sorta jumps out at the reader without much purpose. Hence, I propose putting some indication of the purpose back in the Introduction section.

*Reply: Internal structure is a more accurate description and we will use that.*

p. 8, line 17 – 'ground surface' is a little confusing here. Do you mean where the ice (at depth) is in contact with the ground or do you mean from the surface of the rock glacier, which one must stand on. I suggest being more accurate with this and all other instances of 'ground surface' in the paper.

*Reply: We did mean the surface of the rock glacier that a person would stand on. We will change the terminology to "The presence of clean ice was verified via field excavations, with a sediment-clean ice interface commonly present at < 30 cm depth."*

p. 10, lines 31-32 – the sentence "At this location, the rock glacier is cored by > 10 m of clean ice (Fig. 11b)," – should 'cored' be 'covered'? I didn't think that manual core collection got this deep. . . Oh wait, maybe you aren't meaning this to be the verb that has to do with collecting an ice sample. . . I'd suggest changing 'cored' to 'made up of' or something like that so that there's no confusion.

*Reply: Because we use "core" as a verb (when we cored for ice samples) and a noun (ice-cored landform), we completely agree that this would be confusing. We will only use "core" and "cored" to refer to the ice core samples that we gathered in the field. In all other places we will change the description, such as "At this location, the rock glacier contains >10 m layer of clean buried ice."*

---

## Author Comment (AC2) · 18 Sep 2019

*We thank the anonymous reviewer for their time, expertise, and their insightful and helpful suggestions to make the manuscript more accessible, accurate, and impactful.*

Anonymous Referee #1 Summary:

**Reviewer Comment 1:**
This article looks at a rock glacier in Taylor Valley and concludes that observed glacial successions are useful for studying past climates in the MDV. Overall, I think the article is well described and written, but I do have issues with their age estimate based on pond salts, as well as the description of the pond samples. Unfortunately, this is a key argument in the paper, so the lack of an age constraint requires changes to the discussion and conclusions. However, I don't think this greatly impacts the overall point.
***Reply to Comment 1:***
*We have replied to the reviewer's specific comments about the lake samples and the age estimate below.*

Detailed comments:

**Reviewer Comment 2:**
Figure 2: I know that you can tell what is downhill from the contours, but at first glance the figure seems turned around and was quite confusing, until I realized it was oriented differently from Fig. 1. I recommend including a big bold arrow and 'downhill' text, so that this is immediately clear.
***Reply to Comment 2:***
*We agree that the orientation was probably confusing because North was down.*
***Author changes in manuscript:***
*We will flip the Figure around so that North is up.*

**Reviewer Comment 3:**
Section 3.2: I'm a little confused as to exactly what samples were collected from the ponds, so more clarity is needed. Were all the ponds frozen during sampling? If so, where were the samples taken? From the upper surface, from the margins? I'm particularly curious because some of the pond samples have remarkably high salt concentrations, which is difficult to reconcile with samples of surface ice, which I'd expect to have low concentrations.
***Reply to Comment 3:***
*Thank you for pointing out our lack of detail in describing the lake samples.*
*All but one of the lakes were completely frozen during sampling. All samples were taken from the 3–5 cm depth in the center of the lake ice, after manually removing the upper 3 cm of the ice. One lake, L3, was not frozen solid, but was starting to thaw and to become slushy. However, it was not fully thawed and the person sampling could still walk out onto the lake slush. The ice below 10 cm depth was more solid. We were monitoring air temperatures at the time, because we were also ice coring, and air temperatures were consistently < -8° C. The pond was shaded by the steep hill to the north and so the early surface slush was likely the result of its high salinity and early top-down melting.*

*Author changes in manuscript:*
*In the Methods section, we will include information on pond sampling depths and locations, as well as the information regarding the frozen/slushy nature of the surface ice in pond L3.*

**Reviewer Comment 4:**
Table 3: The authors should fully list all of the ions analyzed and their concentrations, as well as the listed total salt content. Assuming that the isotopes were analyzed for the same samples, these values should also be listed. I also recommend adding in the elevation to the table for easy reference.
*Reply to Comment 4:*
*We do need to include all data with this paper (isotopes and ions). It was a definite oversight that we did not include the data with the original manuscript.*
*Author changes in manuscript:*
*In the Supplementary Data, we will include tables for: (1) ion concentrations for the lake cores, (2) ion concentrations for the lake near-surface samples, and (3) stable isotopic data for all samples: lake, buried ice and glacier ice.*

**Reviewer Comment 5:**
Section 4.3: It's worth mentioning here that samples falling below the local water line suggest either evaporation or sublimation.
*Author changes in manuscript:*
*We will mention evaporation / sublimation earlier in the text at this point as an explanation for stable isotopic data falling below the local meteoric water line, and not just in the Discussion section.*

**Reviewer Comment 6:**
Section 5.3: The major source of ions to these ponds is likely dissolution and aeolian transport of salts from nearby soils, as well as inputs of snow and ice melt, and probably a small component due to direct weathering or atmospheric inputs. See the following references on Taylor Valley salts: Keys, J. R. H. and K. Williams (1981). "Origin of crystalline, cold desert salts in the McMurdo region, Antarctica." Geochimica et Cosmochimica Acta 45(12): 2299-2309. Toner, J. D., et al. (2013). "Soluble salt accumulations in Taylor Valley, Antarctica: Implications for paleolakes and Ross Sea Ice Sheet dynamics." Journal of Geophysical Research 118(1): 198-215. In general, this section needs improvement. Part of the difficulty I'm having with it is that I don't know exactly what was sampled, and if only a surface ice sample was collected, then the measured chemistry would have little relation to the bulk pond composition/salinity. Also, the age analysis using Cl- accumulation is poorly justified. First, only a snowfall source is invoked, but as I mention, this is probably minor relative to Cl- fluxes from surrounding soils. Also, this assumes that the pond is a closed system, so that all the Cl- that goes in does not come out, but this seems unlikely for a pond perched on a valley slope. Throw in the aforementioned uncertainty about how representative the samples are. Basically, I don't think you can come up a with an age using snowfall. I recommend just removing this paragraph. However, given that the age of these ponds is a major point in the article, this would involve some general restructuring of the later discussion and conclusions. On the whole, I find the argument for glacial successions is robust, but not the age estimate based on salt accumulation.

***Reply to Reviewer Comment 6:***
*The age of the ponds is not the point of the article, and we agree with the reviewer that our estimate is rough (as stated in the article) and based on a series of assumptions. If we removed this section entirely, it would not compromise the main conclusions or the scientific impact of the article.*

*As suggested by the reviewer, we will remove the absolute age estimate (40,000 years) from the text. That estimate is based on a series of assumptions and therefore likely has significant error. However, the high salinity of pond L3 does have implications for surface processes and the potential pre-Holocene age for the lower rock glacier. Therefore, we propose to replace the **quantitative** discussion of pond age in sections 5.3 and 5.4 with a **qualitative** discussion.*

*As stated by the reviewer, if there is a soil-weathering source of Cl-, then our calculated age would be an overestimate. Notably, several of the issues raised by the reviewer actually highlight the fact that our age might be an underestimation:*

a. *If the bulk chemistry of the ponds is significantly different from the near-surface ice, this would imply occasionally stratified waters where the saltier waters (ice) are at the bottom. Thus, the surface ion concentrations could be a minimum average composition for the ponds, and our rough age estimate would err toward the young.*

b. *Witherow et al. (2006), examined the ion flux to this part of Taylor Valley, and found that most ion sources were from marine aerosols contained within snow blown in from the coast. We concur that a soil source of major ions, especially Cl-, would decrease the estimated age of L3. However, our conclusion that the lower rock glacier predates the Holocene based on L3 salinity is only compromised if the soil source of Cl- is four times that of the aerosol contribution from snow. This condition also demands that all Cl- is delivered from the soils to the pond and that the pond does not lose ions.*

   *Nearly all local bedrock and rock glacier clasts are granitic, containing low concentrations of Cl-. Small, nearby cinder cones have produced slopes with scoria granules and cobbles, which have much higher concentrations of Cl-. It is certainly possible that some of the Cl- in the ponds originates from these cinder slopes. For instance, we observed two categories of salt crusts at/near the surface of the study area. First, we noted thin layers of salt marking previous extents of existing snowbanks. Second, salt efflorescence is present on the underside of some mafic cinder clasts on the cinder slopes. This supports a case that both snow and wind-blown local sources are possible contributors to the ponds.*

c. *If the Cl- is percolating out of the system, we are again left with a conservative estimate of Cl- accumulation in the depression and therefore of the pond age. Our major ion ratios do not support a strong migration trend: migration of ions during freeze-thaw cycles preferentially concentrates the least soluble ions (e.g., Mg++) in upslope areas, while the more soluble ions (e.g., Na+) travel downslope. However, our Mg++:Na+ is highest in the most downslope, saline pond, L3, indicating that*

*preferential removal of ions is not responsible for the overall trend in salinity. Because the pond does completely dessicate on occasion, windblown loss of ions from surface sediments is a likely process through which Cl- could be removed from the depression and the pond.*

**Author changes in manuscript:**
*In the final manuscript, we propose to revise the Discussion and Conclusion sections as follows:*

1) *We will remove the final paragraph of Section 5.3 (page 11, lines 1–14).*
2) *In place of this paragraph, we will include a discussion of the high salinity of pond L3 and how that might indicate long surface age for the lower rock glacier. We will keep this discussion qualitative rather than providing an absolute age estimate for the pond. This new paragraph will include:*

   (a) *A discussion of the various sources of ions (and specifically Cl-) to central Taylor Valley as outlined by Witherow et al. (2006), Keys & Williams (1981) and Toner et al. (2013).*
   (b) *How ions might be transported to and removed from the pond to highlight what high salinity of L3 might mean for surface processes and pond antiquity. We will include increased wind-blown snow accumulation to the depression, transport of ions to the pond by near-surface water flow, removal of ions by wind especially during dessication events, and the fact that some ions will not be transported to the pond but rather stored in surrounding soils.*
   (c) *We will state that the high salinity of the pond (and possibly of the surrounding soils – unfortunately we did not measure soil salinity) supports the potential antiquity of the lower rock glacier because both aerosol and weathering sources for ions require time to accumulate. These data supplement other data that indicate a long, complex history for the rock glacier (weathering, grussification and GPR analyses).*

3) *We will edit Section 5.4 to remove text that mentions absolute ages based on pond L3 (MIS 3 and MIS 5). We will instead focus on the evidence for multiple glacial advances that are contained in one large rock glacier (1-km long) that is sourced from a relatively small cold-based alpine glacier (3-km long). We will also highlight the potential pre-Holocene antiquity of the lower rock glacier as evidenced by grussification, clast weathering, thick sediment cover, and a high salinity pond in the lower rock glacier, coupled with GPR evidence that supports >4 episodic advances and retreats of the source glacier.*
4) *We will delete the Cl- age estimate from the Conclusions section, which is presently only one sentence. This will not compromise the main Conclusions of the article: (a) the buried ice is sourced from the local alpine glacier, (b) that the rock glacier records multiple glacier advances that likely extend from the present to pre-Holocene, and (c) that rock glaciers and ice-cored drift could be used elsewhere in Antarctica to map and record glacier advance, especially where debris-starved cold-based glacier ice occurs.*

---

## Author Response (AR1)

*We thank the anonymous reviewers for their time, expertise, and their insightful and helpful suggestions to make the manuscript more accessible, accurate, and impactful.*

**Anonymous Referee #1** Summary:

**Comment 1:**
This article looks at a rock glacier in Taylor Valley and concludes that observed glacial successions are useful for studying past climates in the MDV. Overall, I think the article is well described and written, but I do have issues with their age estimate based on pond salts, as well as the description of the pond samples. Unfortunately, this is a key argument in the paper, so the lack of an age constraint requires changes to the discussion and conclusions. However, I don't think this greatly impacts the overall point.
***Reply to Comment 1:***
*We have replied to the reviewer's specific comments about the lake samples and the age estimate below.*

Detailed comments:

**Comment 2:**
Figure 2: I know that you can tell what is downhill from the contours, but at first glance the figure seems turned around and was quite confusing, until I realized it was oriented differently from Fig. 1. I recommend including a big bold arrow and 'downhill' text, so that this is immediately clear.
***Author changes in manuscript:***
*Figure 2: We flipped the orientation of Figure 2 so that North is up.*
*Figure 8: We flipped the orientation of Figure 8 so that North is up.*

**Comment 3:**
Section 3.2: I'm a little confused as to exactly what samples were collected from the ponds, so more clarity is needed. Were all the ponds frozen during sampling? If so, where were the samples taken? From the upper surface, from the margins? I'm particularly curious because some of the pond samples have remarkably high salt concentrations, which is difficult to reconcile with samples of surface ice, which I'd expect to have low concentrations.
***Author changes in manuscript:***
*Page 4, line 23-29: We added information in the Methods section regarding the exact sampling strategy of the buried ice and pond ice hand samples, including depths and location on the ponds.*

**Comment 4:**
Table 3: The authors should fully list all of the ions analyzed and their concentrations, as well as the listed total salt content. Assuming that the isotopes were analyzed for the same samples, these values should also be listed. I also recommend adding in the elevation to the table for easy reference.
***Author changes in manuscript:***
*Supplement: In the Supplementary Data File, we included two tables presenting all of the (1) ion data and (2) stable isotope data for all samples: lake, buried ice and glacier ice.*

**Comment 5:**
Section 4.3: It's worth mentioning here that samples falling below the local water line suggest either evaporation or sublimation.
*Author changes in manuscript:*
*Page 7, line 24-25: We added a sentence on evaporation fractionation.*

**Comment 6:**
Section 5.3: The major source of ions to these ponds is likely dissolution and aeolian transport of salts from nearby soils, as well as inputs of snow and ice melt, and probably a small component due to direct weathering or atmospheric inputs. See the following references on Taylor Valley salts: Keys, J. R. H. and K. Williams (1981). "Origin of crystalline, cold desert salts in the McMurdo region, Antarctica." Geochimica et Cosmochimica Acta 45(12): 2299-2309. Toner, J. D., et al. (2013). "Soluble salt accumulations in Taylor Valley, Antarctica: Implications for paleolakes and Ross Sea Ice Sheet dynamics." Journal of Geophysical Research 118(1): 198-215. In general, this section needs improvement. Part of the difficulty I'm having with it is that I don't know exactly what was sampled, and if only a surface ice sample was collected, then the measured chemistry would have little relation to the bulk pond composition/salinity. Also, the age analysis using Cl- accumulation is poorly justified. First, only a snowfall source is invoked, but as I mention, this is probably minor relative to Cl- fluxes from surrounding soils. Also, this assumes that the pond is a closed system, so that all the Cl- that goes in does not come out, but this seems unlikely for a pond perched on a valley slope. Throw in the aforementioned uncertainty about how representative the samples are. Basically, I don't think you can come up a with an age using snowfall. I recommend just removing this paragraph. However, given that the age of these ponds is a major point in the article, this would involve some general restructuring of the later discussion and conclusions. On the whole, I find the argument for glacial successions is robust, but not the age estimate based on salt accumulation.

*Reply to Comment 6:*
*The age of the ponds is not the point of the article, and we agree with the reviewer that our estimate is rough (as stated in the article) and based on a series of assumptions. If we removed this section entirely, it would not compromise the main conclusions or the scientific impact of the article.*

*As stated by the reviewer, if there is a soil-weathering source of Cl-, then our calculated age would be an overestimate. Notably, several of the issues raised by the reviewer actually highlight the fact that our age might be an underestimation:*

  a. *If the bulk chemistry of the ponds is significantly different from the near-surface ice, this would imply occasionally stratified waters where the saltier waters (ice) are at the bottom. Thus, the surface ion concentrations could be a minimum average composition for the ponds, and our rough age estimate would err toward the young.*

  b. *Witherow et al. (2006), examined the ion flux to this part of Taylor Valley, and found that most ion sources were from marine aerosols contained within snow blown in from the coast. We concur that a soil source of major ions, especially Cl-, would*

*decrease the estimated age of L3. However, our conclusion that the lower rock glacier predates the Holocene based on L3 salinity is only compromised if the soil source of Cl- is four times that of the aerosol contribution from snow. This condition also demands that all Cl- is delivered from the soils to the pond and that the pond does not lose ions.*

*Nearly all local bedrock and rock glacier clasts are granitic, containing low concentrations of Cl-. Small, nearby cinder cones have produced slopes with scoria granules and cobbles, which have much higher concentrations of Cl-. It is certainly possible that some of the Cl- in the ponds originates from these cinder slopes. For instance, we observed two categories of salt crusts at/near the surface of the study area. First, we noted thin layers of salt marking previous extents of existing snowbanks. Second, salt efflorescence is present on the underside of some mafic cinder clasts on the cinder slopes. This supports a case that both snow and wind-blown local sources are possible contributors to the ponds.*

c. *If the Cl- is percolating out of the system, we are again left with a conservative estimate of Cl- accumulation in the depression and therefore of the pond age. Our major ion ratios do not support a strong migration trend: migration of ions during freeze-thaw cycles preferentially concentrates the least soluble ions (e.g., Mg++) in upslope areas, while the more soluble ions (e.g., Na+) travel downslope. However, our Mg++:Na+ is highest in the most downslope, saline pond, L3, indicating that preferential removal of ions is not responsible for the overall trend in salinity. Because the pond does completely dessicate on occasion, windblown loss of ions from surface sediments is a likely process through which Cl- could be removed from the depression and the pond.*

**Author changes in manuscript:**

*Page 11, line 7-26: We rewrote the third paragraph of section 5.3 entirely. The new text is now more conservative in assigning an absolute age to the pond and the text is more thorough in describing the main $Cl^-$ sources and uncertainties outlined by the reviewer and the editor. We decided to keep the 40,000-year estimate in the text as a starting point for a discussion on the main uncertainties that could cause that age to be either an underestimation or an overestimation. And therefore, we no longer state a definitive age for the surface surrounding the pond. We then present the other evidence indicating antiquity of the lower rock glacier.*

*Page 11, line 31: We deleted "associated with MIS 3" and added "pre-Holocene."*

*Page 12, line 1–5: We deleted the text regarding MIS 3 and MIS 5.*

*Page 12, line 21: We deleted the sentence regarding the age of the pond based on $Cl^-$ from the Conclusions.*

**Anonymous Referee #2**

This manuscript details a comprehensive field investigation of a very remote rock glacier in the dry valleys of Antarctica. The authors carefully assess the geomorphology of the rock glacier, examine its internal form using geophysics, and evaluate the chemical and isotopic make up of its ice and those of the meltwater ponds atop the glacier. All of this evidence suggests that the rock glacier has been stable for ~40 ky, which is an impressive amount of time, given the controversy over the dynamics of a possible glacially-dammed lake at the mouth of the valley 5-10 ky ago. One thing that continues to challenge many of the researchers in the dry valleys is how such a lake could exist with so much ice in the area (locally) remaining intact. This set of evidence and conclusion is a terrific contribution to the dry valleys cryospheric literature and provides an excellent example of the wealth of secrets kept in these rock glaciers, especially in Antarctica.

*We thank the reviewer so much for these kind comments.*

Minor suggestions:

**Comment 1**
p. 2, line 21 – I suggest that 'subsurface structure' maybe be changed to 'internal structure', and that the authors add "(using GPR)" or something similar. The reason I mention this is that the GPR part of the Methods section sorta jumps out at the reader without much purpose. Hence, I propose putting some indication of the purpose back in the Introduction section.
*Author changes in manuscript*
*Page 2, line 22-23:* changed to "we examined internal structure using ground-penetrating radar (GPR)."

**Comment 2**
p. 8, line 17 – 'ground surface' is a little confusing here. Do you mean where the ice (at depth) is in contact with the ground or do you mean from the surface of the rock glacier, which one must stand on. I suggest being more accurate with this and all other instances of 'ground surface' in the paper.
*Author changes in manuscript*
*Page 8, 22-23:* changed to "The presence of clean ice was verified via field excavations; the contact between clean buried ice and overlying sediments was commonly encountered at < 30 cm depth."

**Comment 3**
p. 10, lines 31-32 – the sentence "At this location, the rock glacier is cored by > 10 m of clean ice (Fig. 11b)," – should 'cored' be 'covered'? I didn't think that manual core collection got this deep. . . Oh wait, maybe you aren't meaning this to be the verb that has to do with collecting an ice sample. . . I'd suggest changing 'cored' to 'made up of' or something like that so that there's no confusion.
*Author changes in manuscript*
*Page 11, line 3-4:* changed to ". At this location, the rock glacier contains > 10 m of clean buried ice (Fig. 11b),"

**Additional Changes to Manuscript by author**
*Page 1, line 27:* minor changes to wording
*Page 4, line 9:* added "in central Taylor Valley" for clarity.
*Page 4, line 23:* minor changes to wording
*Page 5, line 8–9:* updated the list of ions analyzed based on the fact that 15 ions were measured in core samples, but only the seven major ions were measured in pond hand samples.
*Page 7, line 2-5:* We moved the text "Based on repeat aerial photography …" from the Methods section to the Results section. It was out of place in Methods.
*Page 7, line 9:* added references to the new Supplementary Tables S1 (all ion data).
*Page 7, line 19:* added reference to new Supplementary Table S2 (all stable isotope data).
*Page 12, line 7:* added "Holocene (or older)"
*Page 12, line 14:* spelled out ground-penetrating radar for clarity
*Page 12, line 21:* minor changes to wording
*Page 13, lines 1-16:* Added Data availability, competing interests, financial support and review statement
*Page 13, line 24:* due to text changes, deleted reference Berg et al. (2016)
*Page 15, line 6:* due to text changes, deleted reference Joy et al. (2017)
*Page 16, line 12:* added Toner et al., (2013) reference

[revised manuscript text omitted]

---

## Author Response (AR2)

**Editor comments:**

Dear Dr. Swanger,

Thank you for your revised manuscript. You and your team have done a good job in responding to the reviewers' comments. I only have a few editorial comments that need to be treated, but are easily dealt with and are detailed in markup below.

One question that I do have is whether or not you can say something in your discussion that addresses the comment from Reviewer 2 about the implications of your work with respect to a possible glacially-dammed lake at the mouth of the valley 5~10 ky ago? I'm not personally familiar with the literature for this part of the world, but it would seem to be an important point with respect to the broader significance of this paper. If you feel that you can say something substantive, then please add a sort statement to section 5.4. If this instead opens a can of worms, just leave it alone. It is up to you and your co-authors to decide how to proceed.

Best regards,

Peter

**Author Reply:**

Dr. Morse,

Thank you very much for all of your time and attention in overseeing our manuscript. We have made the edits that are outlined in this pdf.

Our study area occurs at 500–1000 m above sea level, well above the maximum level (~350 m asl) of the glacially-dammed Lake Washburn that filled much of Taylor Valley during the LGM. Our investigation does not have direct implications for Glacial Lake Washburn in terms of occurrence, extent, and/or timing. Therefore, we choose not to include a discussion of the lake in this manuscript.

Sincerely,
Kate Swanger

**Line by Line response to editor comments:**

Page 3, line 9-10: We deleted "Loose" and replace cobble/boulders with "isolated clasts"

Page 3, line 21: deleted "This granitic bedrock cliff is ventifacted."

Page 3, line 30: rewrote the mummified seal section to read: "and the landscape is incised by a network of contraction cracks."

Page 4, line25: second "manually " deleted

Page 4, line 28: rewrote as "we sampled slush-ice from the pond center at 3–5 cm depth."

Page 5, line 8: deleted: "buried ice, ponds and glacier ice"

Page 5, line 9: In reply to the comment: "You show more species in your data tables for these ices so it should be mentioned here," the 14 ion species listed here are the same as those shown in the supplementary table with the complete dataset.

Page 5, lines 21–23: deleted both "approximately"

Page 6, lines 10–12: rewrote section as suggested.

Page 7, lines 24-25: We added this sentence because a reviewer asked us to. We agree that it is a discussion-related topic (and it is also stated in the Discussion), so we deleted it.

Page 8, lines 7–8: rewrote section as "…support a glacial origin for the rock glacier ice"

Page 8, lines 14–16: we made all requested edits.

Page 8, lines 21–24: we made all requested edits.

Page 8, line 28: data changed to "materials"

Page 9, line 10–14: we made all requested edits.

Page 9, lines 32–34: we made all requested edits.

Page 10, lines 2–3: In Figure 10, we added arrows to highlight the debris layers. These layers record periods of glacier retreat. We also added the following text to the figure caption: "Arrows point to reflection events interpreted to be internal debris layers, likely recording retreat of the source glacier."

Page 11, line 26: As stated above our study area occurs at 500–1000 m above sea level, well above the maximum level (~350 m asl) of the glacially-dammed lake(s) that filled much of Taylor Valley during the LGM. Therefore, our study does have direct implications for the lake(s) and we choose not to include a discussion of the lake(s) in the manuscript.

Page 17: Table 1. We changed the table as requested and it is much better. Thank you!

Page 18: Table 2. We changed the table as requested.

Page 19: Table 3. We changed the table as requested and it now includes the data for the seven major ions that are also shown in Figure 8. This is a much better table now. Thank you for the thoughtful suggestion.

Page 29: Figure 10. We added arrows to highlight the debris layers and rewrote the caption for clarity. We added text to the caption to highlight how the clean ice and debris layer structure likely records advance/retreat of the alpine glacier. We also changed the axis labels for consistency.

Page 30: Figure 11. We changed the axis labels for consistency.

[revised manuscript text omitted]

---

## Author Response (AR3)

**Editor comments:**

Dear Authors,

Thank you again for your work. It is almost ready to publish. In addition to some minor technical edits, I would like to see the remaining radargrams added to the Supplement. These images are not needed to accentuate any precise argument, but are required to support some of the general statements. Further, this will provide the reader with the complete GPR data set, as you do for geochemistry. Please indicate in the Supplementary figure captions the antenna used in each case (as you did in Figure 11). You will probably want to refer to these supplementary data within Section 4.4. Please see the attached Comments to Author for details.

**Author Reply:**

Dr. Morse,

Thank you for your thoughtful editing of our manuscript. We have added all of the longitudinal GPR lines (400 MHz) to the supplement along with an altered version of Figure 2 showing the locations and start/end points for the lines.

Sincerely,
Kate Swanger

**Line by Line response to editor comments:**

Page 8, line 3-5: We added the text: "Two selected longitudinal lines are shown in Fig. 11; all five longitudinal 400 MHz lines are provided in the supplementary data."

Tables 1, 2, 3: we have changed meter to "metre" and centimeter to "centimetre."

Page 20, Figure 1, line 4: In the editor's pdf the "Landsat 7 imagery" section was tagged, but there were no comments or edits on the tag, so we are unsure of what to change. But we did change the caption to read "Public domain Landsat 7 imagery acquired on December 18, 1999 courtesy of NASA Goddard Space Flight Center" for clarity.

Page 21, Figure 2: We altered Fig. 2 to only include the GPR lines that are shown in Figs. 10 and 11. We also changed the figure caption to correspond to these alterations.

[revised manuscript text omitted]